# A Nonlinear Viscoelastic Constitutive Model for Solid Propellant with Rate-Dependent Cumulative Damage

**DOI:** 10.3390/ma15175834

**Published:** 2022-08-24

**Authors:** Shenghao Chen, Chunguang Wang, Kaining Zhang, Xuan Lu, Qun Li

**Affiliations:** State Key Laboratory for Strength and Vibration of Mechanical Structure, School of Aerospace, Xi’an Jiaotong University, Xi’an 710049, China

**Keywords:** solid propellant, viscoelasticity, constitutive model, rate-dependent cumulative damage, finite element analysis

## Abstract

Solid propellant is a composite material exhibiting classic nonlinear viscoelastic mechanical characteristic, which is due in a large part to a cumulative damage process caused by the formation and growth of microflaws inside. The standard relaxation tests and uniaxial tension tests under different velocities of hydroxyl-terminated polybutadiene (HTPB) propellant are carried out in this paper, where Digital Image Correlation (DIC) technique is applied to record deformation. The experimental results show that the material mechanical behavior is rate-dependent. It is also observed that the yield stress and failure stress are significantly rate-dependent on the tensile velocity. Based on these experimental results, it can be inferred that the stiffness degradation and damage evolution of HTPB propellant are a rate-dependent processes. Therefore, the damage accumulation of HTPB propellant is considered rate-dependent in this research. In order to describe the mechanical characteristic precisely, a nonlinear viscoelastic constitutive model with rate-dependent cumulative damage is developed. The damage model is developed based on the concept of pseudo strain, in which a Prony series representation of viscoelastic material functions is applied. Besides, a rate-dependent damage variable is introduced into the model through considering the rate-dependent characteristics of cumulative damage process. In addition, a new normalized failure criterion is derived on the basis of the proposed damage model, which is independent of strain-rate after normalization. Finally, it is implemented in commercial finite element software for stress analysis to verify the predictive capacities of the damage model. The accuracy of the constitutive model and failure criterion is validated under uniaxial tensile tests of various strain rates.

## 1. Introduction

Solid rocket propellant is a complex particulate composite material and consists of relatively stiff solid particles embedded in a matrix of soft polymeric material [1,2,3,4,5]. Solid propellant exhibits classic nonlinear viscoelastic response to loading and is subject to property degradation over time [6]. Results from a mass of experiments with micro computed tomography (micro-CT) or scanning electron microscopy (SEM) show that the matrix and the particle/matrix interface are the main source of failure under loading, where microflaws (microcracks and microvoids) will easily initiate and grow [2,3,7,8]. The formation and growth of microflaws are largely believed to be the cause of strong nonlinear stress–strain response of solid propellant [9]. The mechanical behavior of solid propellant has a major influence on the performance of a solid propellant rocket.

Experimental datum show that the mechanical behavior of viscoelastic materials depends on time, temperature, loading rate, and superimposed pressure [2,10,11,12,13]. The attempt to account for all characteristics would result in a very complex constitutive model. Thus, the nonlinear viscoelastic constitutive model with certain features of solid propellant has been studied by many researchers. Characterization of the constitutive response considering damage has been the objective of other related research in the past few decades. Many researchers [12,14,15,16,17,18,19,20] have done a mass of works and developed various damage models based on continuum damage mechanics or thermodynamics theory. Most of these models represent strain-softening behaviors, the effect of strain rate, cyclic loading, or some other issues of various kinds of propellant, but they are complicated and would require a large number of experiments to determine model parameters. On the other hand, there is also some research [10,12,20,21,22,23,24] about the implementation of damage models in the finite element method (FEM). In this research, the models are established or modified and extended to FEM, and finally validated by experiments. It is pointed out that some studies have found that the damage behavior of some viscoelastic material is rate-dependent. Park [25] finds that damage growth of asphalt concrete exhibits rate dependence, which is affected by multiple aspects. Shunmugasamy and Gupta [26] find strain rate dependence of syntactic foam properties and failure mechanisms using micro-CT scan to evaluate the damage profile. Kothari et al. [27] develop a rate-dependent damage model of polymer networks, and they consider damage events as the breakage of the bond, which are rate-dependent processes.

There is also some research about the damage behavior of HTPB propellant [12,24,28,29]. However, few models exist in the literature expressing the rate-dependent characteristic of damage growth explicitly for solid propellant. In this research, a series of uniaxial tension tests under different tensile velocities of HTPB propellant are carried out, and deformation information is recorded by DIC technique. The yield stress and failure stress are observed experimentally and related to the tensile velocity, which indicates that the stiffness degradation and damage evolution process of HTPB propellant are rate-dependent. It is inferred that the damage accumulation is rate-dependent based on the above analysis. Therefore, a rate-dependent damage model describing the characteristic is necessary to be established. In the present study, such a damage model is presented, where a rate-dependent damage variable is introduced to represent the rate-dependent characteristic of cumulative damage process. Because the failure stress and the damage variable are rate-dependent, it is difficult to describe the damage degree of propellant. Considering these issues, a new failure criterion is derived after normalization, which is independent of strain-rate. The numerical simulation stress response and prediction of failure give a high agreement with experimental data.

## 2. Constitutive Model

### 2.1. Rate-Dependent Damage Model

The proposed constitutive model in this work is evolved on the basis of Schapery’s viscoelastic constitutive theory [17]. The constitutive theory for viscoelastic material with growing damage and other changes in structure is developed by replacing the strain *ε* in an elastic formulation with pseudo strain *ε^R^* defined by Equation (1)
(1)$$\varepsilon_{}^{R}=\frac{1}{E_{R}}\int_{0}^{t}E\left(t-\tau \right)\frac{\partial \varepsilon }{\partial \tau }d\tau$$
where, *E*(*t*) is relaxation modulus, *t* is reduced time, *E_R_* is reference elastic modulus which has the same dimension as the relaxation modulus, and the time argument *τ* is specified as the variable of integration.

The initiation and growth of microcracks and microvoids occur under loading. During this process, damage behavior primarily appears, which leads to a nonlinear material behavior. In order to describe the nonlinear characteristic accurately, a viscoelastic constitutive theory is developed for particulate composites with growing damage, which is based on thermodynamics of irreversible process with internal state variables [12]. The internal state variables, *S_m_*(*m* = 1, 2, …, *M*), are independent variables, which serve to account for the effects of damage and other microstructure changes. In this theory, stress–strain relation gives that if only one internal variable is selected
(2)$$\sigma =\frac{\partial W^{R}}{\partial \varepsilon^{R}}=C\left(S\right)E_{R}\varepsilon^{R}$$
where, *σ* is the stress tensor. *C*(*S*) is the soften parameter, which is a function of *S* and used to describe the nonlinear property of viscoelastic material. *W^R^* is pseudo strain energy density and the expression gives that
(3)$$W^{R}=\frac{1}{2}E_{R}C\left(S\right){\left(\varepsilon^{R}\right)}^{2}$$

As previously discussed, the determination of damage evolution equation is essential to accurately describe the mechanical behavior of propellant. The formation and growth of microvoids and microcracks constitutive a cumulative damage process [29]. Considering that propellant is a classical viscoelastic material, Duncan [29] proposed a cumulative damage function, which is defined as a function of stress–time loading history, is given by
(4)$$S=k\int_{0}^{t}\sigma {\left(t\right)}^{\beta }dt$$
where *k* and *β* are characteristic cumulative damage parameters of the material. They can be obtained from tensile test results at different strain rates.

The advantage of the approach is that it enables the damage evolution in the viscoelastic material to be related to loading history. However, it is observed experimentally in this research that the damage accumulation of propellant exhibits rate dependence (see in Section 3.1). Hence, we consider that the parameter *k* above is no longer seen as a constant, but a function of strain rate, *k*($\dot{\varepsilon }$). In this research, it is assumed that the form of *k*($\dot{\varepsilon }$) is the same as the stress–time loading history part of Equation (4) for simplicity. That is to say that *k*($\dot{\varepsilon }$) is the power law strain rate function, defined as *k*($\dot{\varepsilon })=a\dot{\varepsilon }$*^α^*. Thus, the proposed rate-dependent damage evolution model is derived as
(5)$$S=a{\dot{\varepsilon }}^{\alpha }\int_{0}^{t}\sigma {\left(t\right)}^{\beta }dt$$
where *a*, *α*, and *β* are model parameters determined experimentally.

Thus, a new nonlinear viscoelastic constitutive model with rate-dependent cumulative damage is established, consisting of Equations (1), (2) and (5).

### 2.2. Relaxation Modulus Function

It can be seen from the established constitutive model that the fundamental behavior of viscoelastic material depends on the pseudo strain. The pseudo strain is dependent on the relaxation modulus. Hence, obtaining the exact relaxation modulus is the first and important step to acquire constitutive model parameters. 

In the case of constant velocity uniaxial tensile test, the integral representation of linear viscoelastic constitutive equation takes the form as Equation (6). A ramp-dependent method [30] is adopted in this research to acquire relaxation modulus, which is expressed as
(6)$$\sigma \left(t\right)=\int_{0}^{t}E\left(t-\tau \right)\dot{\varepsilon }\left(\tau \right)d\tau$$

For viscoelastic materials, stress relaxation is the observed decrease in stress in response to constant strain, as shown in Figure 1.

Relaxation test is the most common approach to determine relaxation properties of materials, in which an initial strain *ε*_0_ is suddenly applied to a specimen and then held constant for a long time. It takes the ramp time, *t*_1_, to reach the initial strain with the constant rate of strain $\dot{\varepsilon_{0}}$ in the relaxation test. The strain in the relaxation test is shown in Figure 2 and it can be written as
(7)$$\varepsilon \left(t\right)=\left\{ \begin{array}{ll} {\dot{\varepsilon }}_{0}t & t<t_{1}\\ \varepsilon_{0} & t\geq t_{1} \end{array}\right.$$

For viscoelastic materials like solid propellant, the rate of relaxation is fast, or the ramp time is large [31]. Hence, considering the effect of ramp stage, it is necessary to acquire the precise relaxation modulus. Substituting Equation (7) to Equation (6), give that
(8)$$\sigma \left(t\right)=\left\{ \begin{array}{ll} {\dot{\varepsilon }}_{0}\int_{0}^{t}E\left(t-\tau \right)d\tau & t<t_{1}\\ {\dot{\varepsilon }}_{0}\int_{0}^{t_{1}}E\left(t-\tau \right)d\tau & t\geq t_{1} \end{array}\right.$$

The stress at time $t\geq t_{1}$ is given by
(9)$$\sigma \left(t\right)={\dot{\varepsilon }}_{0}\int_{0}^{t_{1}}E\left(t-\tau \right)d\tau$$

Then differentiate Equation (9) with respect to time. This yields
(10)$$\begin{array}{ll} \dot{\sigma }\left(t\right) & ={\dot{\varepsilon }}_{0}\int_{0}^{t_{1}}\partial_{t}E\left(t-\tau \right)d\tau =-{\dot{\varepsilon }}_{0}\int_{0}^{t_{1}}\partial_{\tau }E\left(t-\tau \right)d\tau \\ & ={\dot{\varepsilon }}_{0}\left(E\left(t\right)-E\left(t-t_{1}\right)\right) \end{array}$$

Then the relaxation modulus at time *t* is given by
(11)$$E\left(t\right)=\frac{\dot{\sigma }\left(t\right)}{{\dot{\varepsilon }}_{0}}+E\left(t-t_{1}\right)$$

Using two point trapezoidal rule to integrate Equation (9) numerically gives
(12)$$\sigma \left(t\right)=\frac{1}{2}{\dot{\varepsilon }}_{0}t_{1}\left(E\left(t-t_{1}\right)+E\left(t\right)\right)=\frac{1}{2}\varepsilon_{0}\left(E\left(t-t_{1}\right)+E\left(t\right)\right)$$

Substituting Equation (11) in Equation (12) yields
(13)$$E\left(t-t_{1}\right)=\frac{\sigma \left(t\right)}{\varepsilon_{0}}-\frac{\dot{\sigma }\left(t\right)}{2{\dot{\varepsilon }}_{0}}\;\;t\geq t_{1}$$

Or
(14)$$E\left(t\right)=\frac{\sigma \left(t+t_{1}\right)}{\varepsilon_{0}}-\frac{\dot{\sigma }\left(t+t_{1}\right)}{2{\dot{\varepsilon }}_{0}}\;\;t\geq 0$$
where ${\dot{\varepsilon }}_{0}=\varepsilon_{0}/t_{1}$. The stress rate is obtained by numerical differentiation.
(15)$$\dot{\sigma }\left(t\right)=\frac{\sigma \left(t+h\right)-\sigma \left(t-h\right)}{2h}$$
where *h* is the length of the time step.

For a typical viscoelastic material, the relaxation modulus can be represented by a power law, that is in the form of a Prony series. In this paper, we use the Prony series form of Equation (16) to express the relaxation modulus [32].
(16)$$E\left(t\right)=E_{\infty }+\sum_{i=1}^{n}E_{i}e^{-\frac{t}{\tau_{i}}}$$
where, *E_∞_*, *E_i_* are equilibrium relaxation modulus coefficient and i-th relaxation modulus coefficient respectively, *τ_i_* is *i*-th reduced time coefficient. Combining with the above formulas and relaxation test data, each coefficient of Equation (16) can be obtained by the Nonlinear Least Squares Method, which is implemented by the fit function of MATLAB software.

### 2.3. Constitutive Model Parameters Determination

The constitutive model parameters will be acquired by different constant velocities uniaxial tensile experiments, including a specific form of softening function *C*(*S*) and damage parameters *a*, *α*, *β*. It is assumed that *C*-*S* curves of different strain rates will be overlapped at constant temperature in this paper. The detailed description of specific fitting process of constitutive parameters is as follows:(1)It is clear from Equations (1) and (2) that *E_R_* will be eliminated during the calculation of stress. Without loss of generality, select reference modulus *E_R_* = 1 and take the Prony series expression Equation (16) for relaxation modulus into Equation (1) to obtain the pseudo strain *ε^R^*. That is to say that the *ε^R^*-*t* curves can be obtained.(2)According to the constant velocity tensile experimental results, the *σ*-*t* curves can be obtained. Combining the *ε^R^*-*t* curves, the *C*-*ε^R^* curves can be obtained by Equation (2).(3)Assign initial values to damage parameters (*a*, *α*, *β*) and damage internal variable *S* can be calculated by Equation (5). Next, *S*-*ε^R^* relationship can be obtained, and *C*-*S* relationship can be ensured.(4)Plot all *C*-*S* curves under different strain rates together and determine whether the overlap ratio is good enough. Then, the values of damage parameters (*a*, *α*, *β*) will be adjusted appropriately until the *C*-*S* curves have a good contact ratio, and the damage parameters are what we want.(5)According to the final *C*-*S* curves, determine the form of softening function *C*(*S*) and the values of the parameters in the function.

The fitting procedure is summarized in Figure 3 as a flowchart chart.

## 3. Constitutive Model Calibration

### 3.1. Uniaxial Tensile Tests and Relaxation Tests of HTPB

The material studied in this paper is HTPB propellant, which is produced by curing at 60 ℃ for seven days. The main chemical components include HTPB, AP (ammonium perchlorate), AL (aluminum powder), and the others. The mass fractions are 8%, 69.5%, 18.5%, and 4%, respectively. The specimens and chucks were designed according to the aerospace industry standard of PRC, QJ 924-1985, as illustrated in Figure 4. To obtain the HTPB tensile mechanical properties under different strain rates, stress relaxation tests and constant velocity tensile tests were conducted. In the relaxation test, specimens were stretched at a tensile velocity of 100 mm/min to strain level of 5% and kept the strain constant for 2000 s according to the aerospace industry standard of PRC, QJ 2487-1993. As for the constant velocity tensile experiments, 1, 10, 20, and 100 mm/min were selected as the tensile velocities, and the corresponding strain rates were 0.00021, 0.0021, 0.0042, and 0.021 s^−1^. Each tensile experiment was carried out until the sample broke. Each test was repeated at least 3 times, and the final results were averaged. The standard deviations for averaged values are within 2%. The constant velocity tensile experiments were carried out in accordance with the aerospace industry standard of PRC, QJ 924-1985. Experimental facilities included Zwick/Roell-Z005 type universal testing machine, DIC equipment (VIC-3D, Beijing Ruituo Technology Co., Ltd., Beijing, China), and two cameras with a resolution of 2048 × 2048, with 50 mm macro lens attached. The schematic diagram of experimental equipment is shown in Figure 5.

DIC is a non-contact deformation measurement technique to measure full-field strain over the surface of specimen [33]. What is more, the technique has also been applied to study the viscoelastic behavior of HTPB [34]. The basic principle is to compare the position of pixels in the original and deformed images. In order to increase accuracy of measurement, speckle patterns are usually applied by spraying the specimen with contrasting paints (i.e., black, white, and gray). The surface information of HTPB was quantified using the software VIC-3D [35]. A standard calibration target that had 14 dots × 10 dots, with a dot spacing of 14 mm, was used during the calibration process. The strain field in tensile direction within a certain range of specimen is shown in Figure 6a.

The stress–strain curves at different tensile velocities are shown in Figure 7. It can be seen from the figure that the stress responses of propellant are rate-dependent and the modulus and stress responses become larger with the tensile velocity increasing.

Figure 7 shows that the stress–strain relation is nonlinear. In this research, the yield point is determined by the elastic limit point. The dominant stage of elastic strain is from the initial point to yield point, and the stress corresponding to this point is yield stress, denoted as *σ_s_*. In addition, the growth rate of stress gradually decreases to the maximum stress, namely the failure stress. Similarly, the stress at failure point is expressed as *σ_m_*.

The yield stress and failure stress are important parameters describing failure behavior of materials. In order to study the rate dependence of yield stress and failure stress, the curve of stress against the logarithm of strain rates is drawn, as shown in Figure 8. Figure 8 presents the linear relationship between *σ_s_* or *σ_m_* and logarithm of strain rates, which illustrates that *σ_s_* and *σ_m_* are rate-dependent. This indicates that the stiffness degradation and softening behavior of HTPB propellant are obviously affected by strain rates. That is to say that the damage evolution is rate-dependent. It can be concluded that the damage accumulation inside the material is related to strain rates.

### 3.2. Numerical Simulation of Uniaxial Tension

In order to verify the predictive capability of the proposed model, the finite element method is used to carry out relevant numerical simulation and analysis. The constitutive model is implanted in the commercial finite element software ABAQUS for stress analysis by a user subroutine (called UMAT). The subroutine can be used to define the mechanical constitutive behavior of a material. In this section, stress responses and damage variable changes of specimen under four different constant strain rates are calculated.

The model parameters should be determined before the finite element analysis. Based on the stress–time data of relaxation test, the stress rate can be obtained by Equation (15). Substitute the stress rate into Equation (14) to obtain the value of relaxation modulus. The expression of relaxation modulus can be obtained by the Nonlinear Least Squares Method, which is implemented by MATLAB software. The Prony series of relaxation modulus is expressed as
(17)$$\begin{array}{ll} E\left(t\right) & =1.45+0.528e^{-\frac{t}{478.378}}+0.661e^{-\frac{t}{30.727}}+0.699e^{-\frac{t}{4.219}}\\ & +0.724e^{-\frac{t}{1.110}}+0.921e^{-\frac{t}{0.398}}+2.308e^{-\frac{t}{0.104}} \end{array}$$

Following the flow chart as shown in Figure 3, the fitting curve, which is expressed by solid lines, is as shown in Figure 9. The specific rate-dependent damage model parameters can be obtained, give that
(18)$$\left\{ \begin{array}{l} \varepsilon^{R}=\frac{1}{E_{R}}\int_{0}^{t}E\left(t-\tau \right)\frac{\partial \varepsilon }{\partial \tau }d\tau \\ \sigma =C\left(S\right)\varepsilon^{R}\\ S=194.01{\dot{\varepsilon }}^{0.55}\int_{0}^{t}\sigma {\left(t\right)}^{7.60}dt\\ C\left(S\right)=1+10.44S^{0.72}-10.80S^{0.70} \end{array}\right.$$

The *C*-*S* fitting curve is shown as

The finite element model is shown in Figure 10 and its dimension is 50 mm × 10 mm × 10 mm. The specimen is simplified to a rectangular geometry based on the assumption that the stress in the cross section is uniformly distributed under tensile loading. One end of the model is fixed, while the other end is applied for displacement loads in the tensile direction. The finite element mesh applied in this case consists of eight-node 3D stress elements with reduced integration. The mesh size is controlled by a global size of 0.5 mm, and the number of elements is 40,000. One element in the middle of the sample is selected as the research object to obtain the *σ*-*ε* curves and *S*-*ε* curves under different tensile velocities. Figure 11 shows the comparison between predicted results and experimental data. It is shown that the agreement with experimental data is generally well at different strain rates. In order to directly describe the prediction accuracy, the stress values of 10 sample points from the starting point to the failure point of each curve are taken to calculate the root mean square errors (*RMSE*), calculated by Equation (20). As shown in Table 1, *RMSE* value of different tensile rates are all within 2%. It can be concluded that the proposed constitutive model can predict stress responses of HTPB under uniaxial tensile tests.
(19)$$RMSE=\sqrt{\frac{1}{n}\sum_{i=1}^{n}{\left(\sigma_{\exp}-\sigma_{\mathrm{FEM}}\right)}^{2}}$$
where *σ*_exp_ and *σ*_FEM_ are the experimental and predicted stress value of sample points, respectively.

### 3.3. Numerical Validation of Damage Behavior

In order to preliminarily evaluate the capability of the rate-dependent damage model to describe damage behavior of HTPB propellant, the comparison between the experimental results and the predicted value of *σ_s_* and *σ_m_* is shown in Figure 12. As shown in Figure 12, the values of experimental and simulation results are generally close. It can be concluded that the rate-dependent damage model can predict the value of yield stress and failure stress.

In order to analyze the accumulation of damage inside the material, the stress and damage variable against strain under different tensile velocities are plotted, as shown in Figure 13. When the strain is in a small range, the damage accumulation inside the material can be ignored, and the value of damage variable *S* is also small.

When the stress value reaches the value of failure stress *σ_m_* at failure point, the material is considered to be completely damaged in this paper. For convenience, the value of damage variable *S* at this time is in terms of *S_m_*. Figure 14 shows the relationship between *S_m_* and strain rate is linear, and the specific fitting formula is shown in Equation (21).
(20)$$S_{m}=163.6\dot{\varepsilon }+0.39$$

Although the failure stress and damage variable *S* can describe the damage characteristics of HTPB propellant to some extent, their values are both related to strain rates so that they are not suitable as general damage measures. Therefore, defining another rate-independent variable to describe damage degree of the material is necessary. In the research, a new failure criterion is introduced as a damage measure, which is defined as the ratio of *S* and *S_m_*, denoted as $\bar{S}$. The expression of $\bar{S}$ is given as
(21)$$\bar{S}=\frac{S}{S_{m}}=\frac{S}{163.6\dot{\varepsilon }+0.39}$$

The advantage of this approach is that $\bar{S}$ value at different strain-rates is within 1 after normalization, which means that $\bar{S}$ is independent of strain rate. Figure 15 shows the evolution of damage degree as it relates to the experimental data at different tensile velocities. As shown in Figure 15, the condition $\bar{S}$ = 0 defines a material state wherein there is no damage, and the condition $\bar{S}$ = 1 defines the state of failure in the material with the maximum stress attained. Then, $\bar{S}$ is defined as the quantity measuring the damage degree of the material under different strain rates. Therefore, $\bar{S}$ can be used as failure criterion to measure whether the material is at failure.

## 4. Conclusions

In this work, a nonlinear viscoelastic constitutive model with rate-dependent cumulative damage is proposed based on experimental observations, and damage behavior is analyzed based on this model. The main conclusions can be summarized as follows:(1)DIC technique is applied in the relaxation tests and uniaxial tensile tests of HTPB propellant. It is observed experimentally that the value of yield stress and failure stress are rate-dependent, and it is deduced that the damage accumulation is rate-dependent.(2)Based on the experimental results, a rate-dependent damage model of solid propellant is developed through introducing the concept of pseudo strain and softening function. The Prony series is applied to represent relaxation modulus, and a rate-dependent damage variable is introduced to represent the rate-dependent characteristic of damage accumulation.(3)The accuracy of the rate-dependent damage model is verified through the comparison between finite element analysis results and experimental results. The results show that the predictions agree well with the experimental results under different strain rates.(4)Based on the proposed model, a new failure criterion for HTPB propellant is proposed after the damage variable is normalized, which is independent of strain rate. It is hopeful that the proposed failure criterion can provide an effective and available method for the prediction of damage behavior of solid propellant.

## Figures and Tables

**Figure 1 materials-15-05834-f001:**
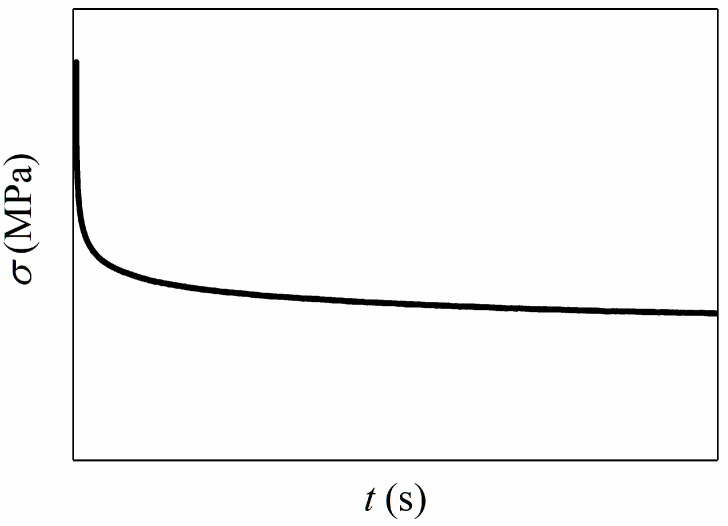
Schematic diagram of relaxation stress response.

**Figure 2 materials-15-05834-f002:**
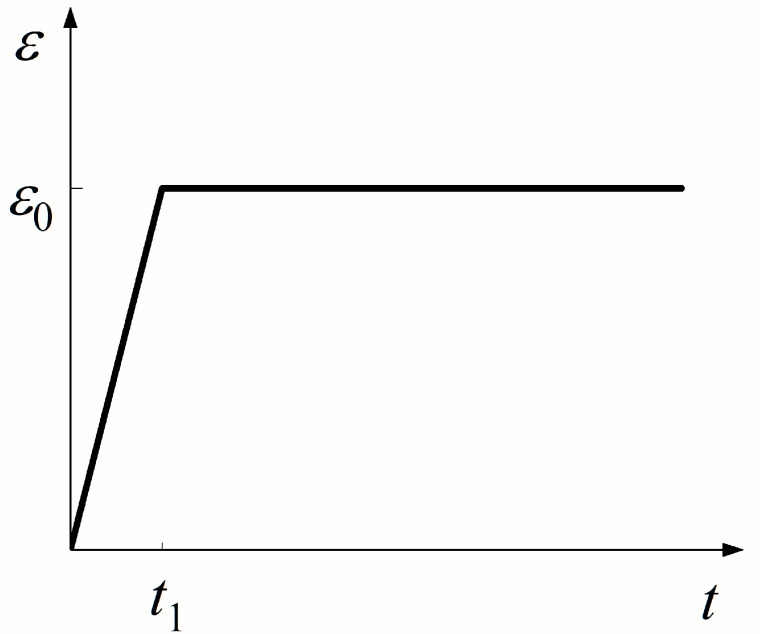
Strain in the relaxation test.

**Figure 3 materials-15-05834-f003:**
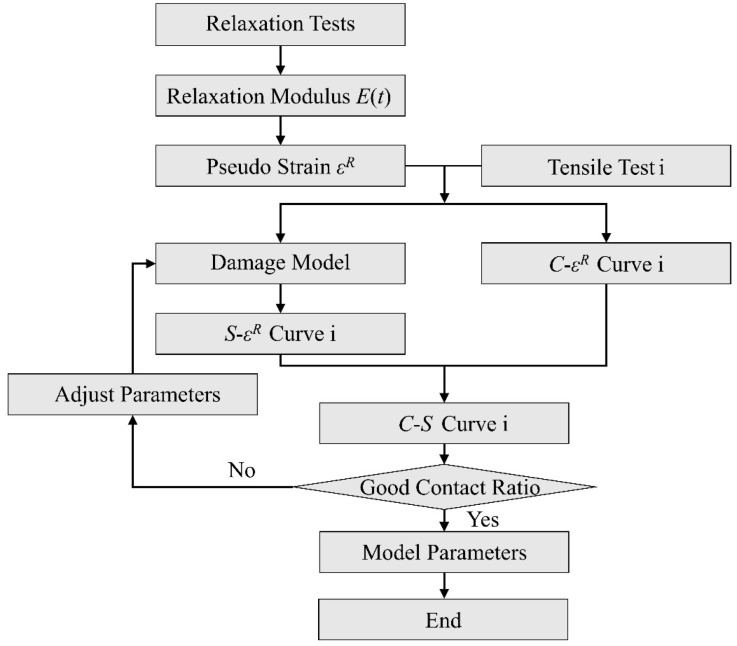
The flow chart of the damage model parameters determination.

**Figure 4 materials-15-05834-f004:**
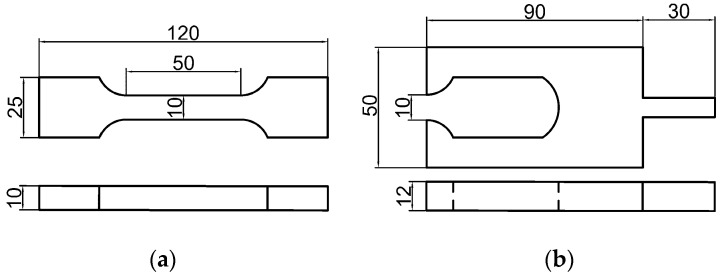
Schematic diagram of specimen and chuck (unit: mm). (**a**) specimen; (**b**) chuck.

**Figure 5 materials-15-05834-f005:**
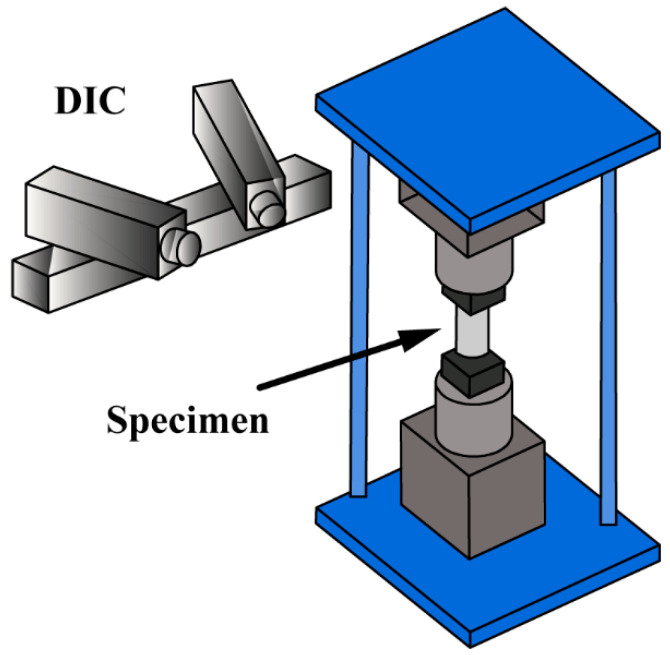
Schematic diagram of experimental equipment.

**Figure 6 materials-15-05834-f006:**
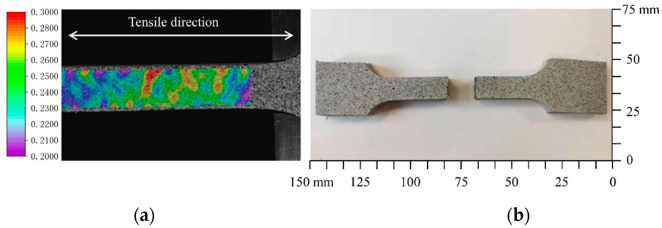
Strain field of specimen and a photograph of broken specimen. (**a**) Strain field of specimen; (**b**) broken specimen.

**Figure 7 materials-15-05834-f007:**
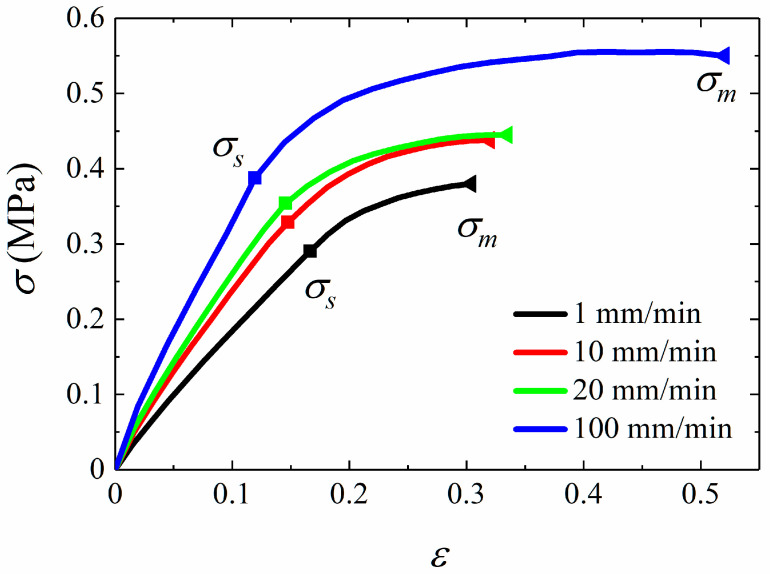
Stress–strain curves of tensile experiments.

**Figure 8 materials-15-05834-f008:**
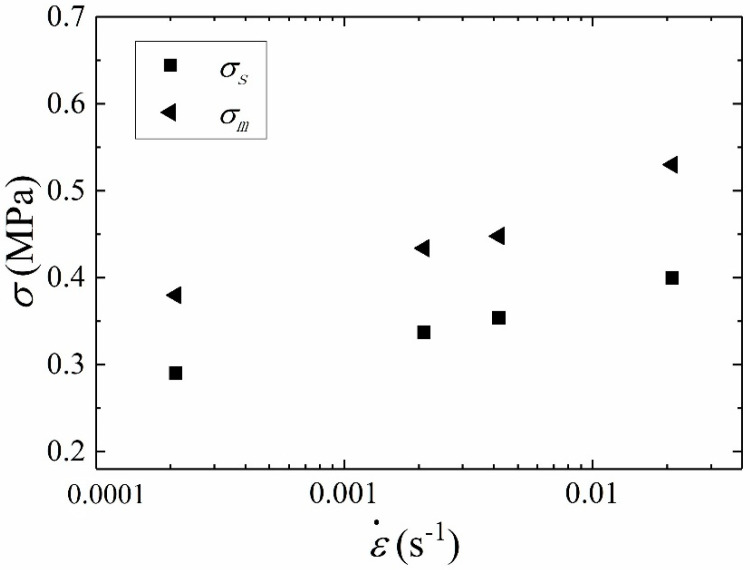
Yield stress *σ_s_* and failure stress *σ_m_* against strain rates.

**Figure 9 materials-15-05834-f009:**
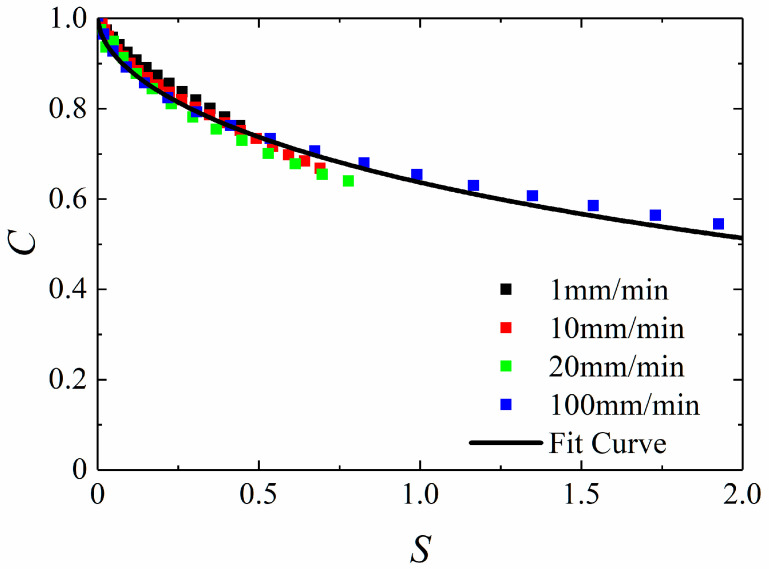
C−S curve of HTPB propellant.

**Figure 10 materials-15-05834-f010:**
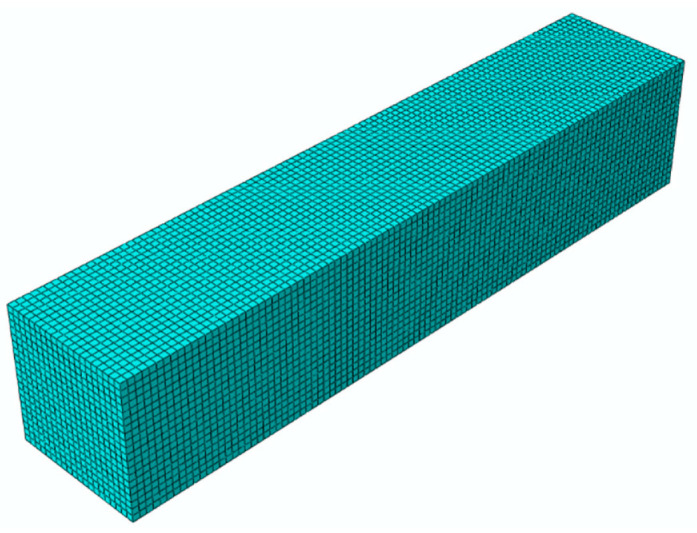
Finite element model in ABAQUS.

**Figure 11 materials-15-05834-f011:**
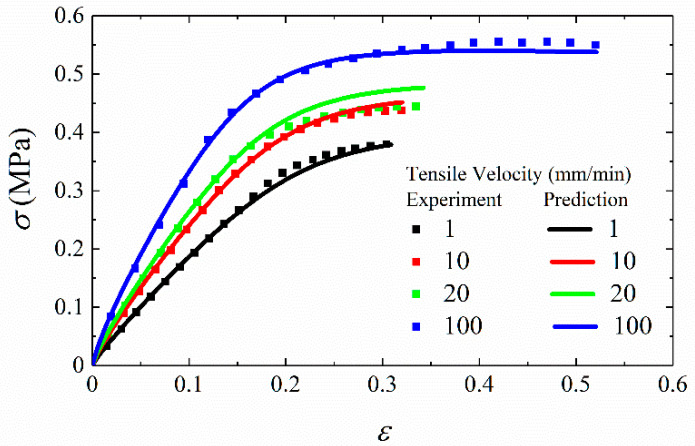
The validation of uniaxial constant strain rate tensile experiments.

**Figure 12 materials-15-05834-f012:**
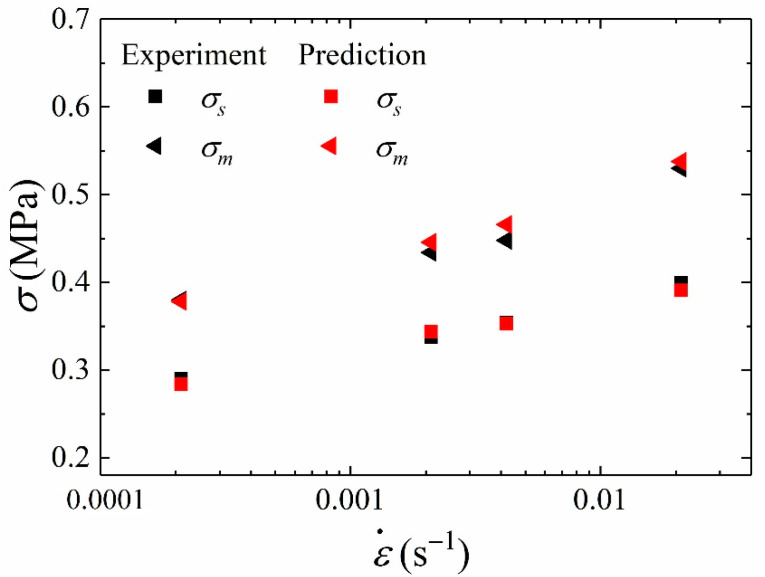
The validation of $\sigma_{m}$ and $\sigma_{s}$.

**Figure 13 materials-15-05834-f013:**
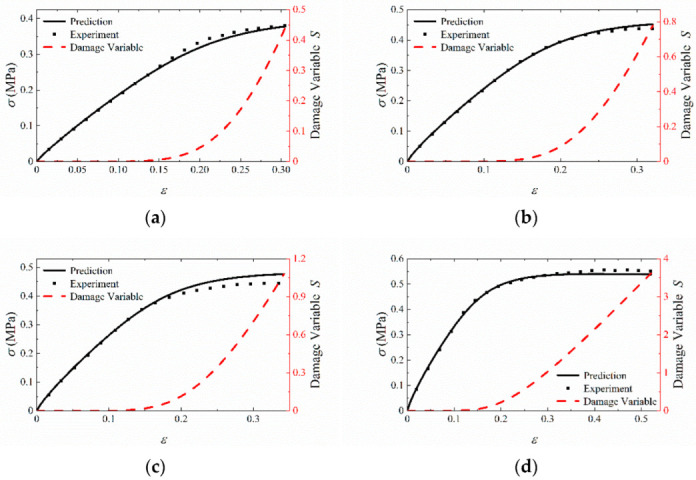
Uniaxial constant strain rate simulation. (**a**) 1 mm/min; (**b**) 10 mm/min; (**c**) 20 mm/min; (**d**) 100 mm/min.

**Figure 14 materials-15-05834-f014:**
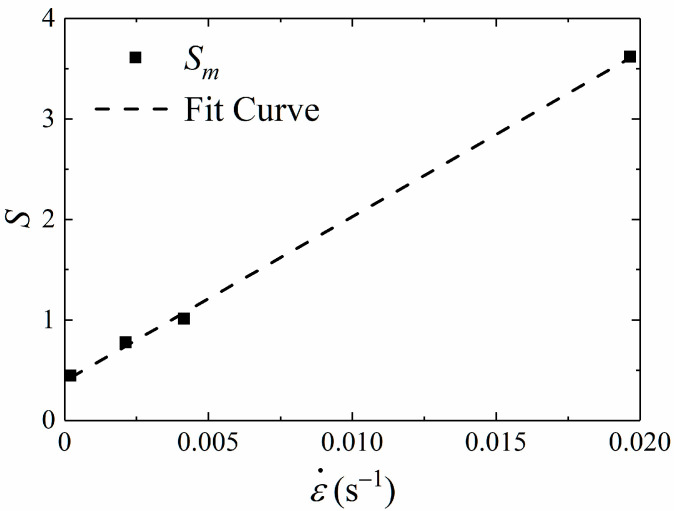
$S_{m}$ against strain rate $\dot{\varepsilon }$ curve.

**Figure 15 materials-15-05834-f015:**
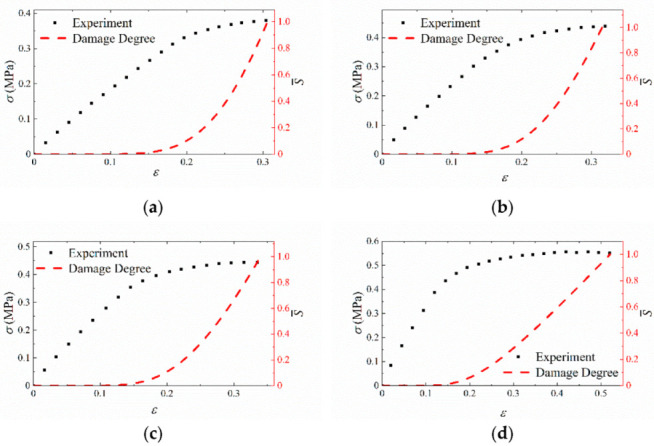
The validation of damage degree $\bar{S}$. (**a**) 1 mm/min; (**b**) 10 mm/min; (**c**) 20 mm/min; (**d**) 100 mm/min.

**Table 1 materials-15-05834-t001:** *RMSE* value of experimental and predicted results under different tensile velocities.

Tensile Velocity (mm/min)	1	10	20	100
*RMSE* (%)	0.73	0.51	1.52	0.90

## Data Availability

Not applicable.

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
