# Peer review of "A Nonlinear Viscoelastic Constitutive Model for Solid Propellant with Rate-Dependent Cumulative Damage"

_materials, 2022, doi:10.3390/ma15175834_

Round 1
Reviewer 1 Report
The authors performed experiments and finite element modeling of hydroxyl-terminated polybutadiene (HTPB) propellants to establish a failure criterion that quantifies the degree of damage in a way that is independent of strain rate. Stress relaxation and constant velocity tensile experiments were performed to calibrate the parameters of the nonlinear viscoelastic model that was developed to study cumulative damage. The finite element simulation modeled a uniaxial tensile test. The theoretical framework for establishing the nonlinear viscoelastic model was provided, along with graphs comparing the results from the simulations with the experiments. The results of this work will be beneficial for those interested in characterizing the nonlinear behavior of composite energetic materials. However, there are a few general comments that need to be addressed:
· More details are needed for the experiments and how the constants were calibrated during the fitting process.
· Throughout the paper, make sure to correct small grammatical errors.
· Expand the Section titled "3.1. Uniaxial tensile tests and relaxation tests of HTPB " by including details of the DIC experiments. Details should include (i) the cameras and lens were used, the (ii) calibration grid size, (iii) the software used for DIC, and (iv) other items detailed in the DIC guide found at https://idics.org/guide/. Make sure to cite the guide.
· The authors should include a figure (or table) that applies Equation 21 so the reader can see the damage degree of the samples from the experiments performed in this study. A statement of how the damage degree from Equation 21 was validated experimentally should be included (e.g. via CT or SEM).
Detailed and textual comments are as follows:
· Pg.1 - Line26, Abstract: Change "criterion is verified" to "criterion is validated".
· Pg.1 - Line 31, 1.Introduction: Change "material consists" to "material and consists".
· Pg.1 - Line 34, 1.Introduction: Define "CT" before using the acronym for computed tomography.
· Pg.1 - Line 40, 1.Introduction: Change "of solid" to "of a solid".
· Pg.1 - Line 41, 1.Introduction: Change "of solid" to "of a solid".
· Pg.2 - Line 54, 1.Introduction: Change "method(FEM)" to "method (FEM)", by adding a space after "method".
· Pg.2 - Line 55, 1.Introduction: Change "verified" to "validated".
· Pg.2 - Line 60, 1.Introduction: Change "micro CT" to "micro-CT" with the hypen to be consistent with the notation from Line 34.
· Pg.2 - Line 68, 1.Introduction: Change "to related" to "and related".
· Pg.2 - Line 68, 1.Introduction: Delete the word "obviously".
· Pg.2 - Lines 79-80, 1.Introduction: Reword the sentence that starts with "Section 2 ...", for clarity.
· Pg.2 - Line 79, 1.Introduction: Change "remaining" to "remainder".
· Pg.2 - Line 91, 2.Constitutive model: Define the variable tau in Equation 1.
· Pg.3 - Line 94, 2.Constitutive model: Change "appear to be primarily" to "appear primarily".
· Pg.3 - Line 101, 2.Constitutive model: Define the variable sigma (stress) in Equation 2.
· Pg.3 - Line 105, 2.Constitutive model: Change "describe mechanical" to "describe the mechanical".
· Pg.3 - Line 107, 2.Constitutive model: Change "is classical" to "is a classical".
· Pg.3 - Line 112, 2.Constitutive model: Change "dependence(see" to "dependence (see" by adding a space before the open parenthesis.
· Pg.3 - Lines 114-115, 2.Constitutive model: Change "is same as stress-time" to "is the same as the stress-time".
· Pg.3 - Line 115, 2.Constitutive model: Change "is power" to "is the power".
· Pg.3 - Line 116, 2.Constitutive model: Change "function, as" to "function, defined as".
· Pg.3 - Line 128, 2.Constitutive model: Change "is in detailed in" to "is detailed in".
· Pg.4 - Line 131, 2.Constitutive model: Add another sentence describing the material and experiment conditions used to generate Figure 1. Otherwise, remove the values on the horizonal and vertical axis in Figure 1 to make the plot generic like Figure 2 is generic.
· Pg.5 - Lines 141-142, 2.Constitutive model: Add a citation to support the statement that "the ramp time is large."
· Pg.5 - Line 142, 2.Constitutive model: Change "acquire precise" to "acquire the precise".
· Pg.5 - Line 143, 2.Constitutive model: Change "Substitute" to "Substituting".
· Pg.5 - Line 153, 2.Constitutive model: Change "is the form of Prony" to "is in the form of a Prony".
· Pg.5 - Line 157, 2.Constitutive model: Provide more details for how the constants from Equation 16 were obtained. For instance, what program was used and what function in the program was used for optimizing the constants?
· Pg.7 - Line 185, 3.Constitutive model calibration: Provide more details on how the HTPB propellant was made (e.g. curing temperature and curing time, type of oxidizer, weight fraction of the components, etc.).
· Pg.7 - Line 187, 3.Constitutive model calibration: Change "tensile mechanic properties" to "tensile mechanical properties".
· Pg.7 - Line 189, 3.Constitutive model calibration: Change "100mm/min" to "100 mm/min", by adding a space between the value and its unit.
· Pg.7 - Lines 189-190, 3.Constitutive model calibration: Add another sentence or two explaining why the chosen tensile velocity of 100 mm/min and strain level of 5% was used for the stress relaxation experiments. 5% strain is small and is not expected to cause large deformations in the sample resulting in the need for a nonlinear viscoelastic model.
· Pg.7 - Line 190, 3.Constitutive model calibration: Change "2000s" to "2000 s", by adding a space between the value and its unit.
· Pg.7 - Line 190, 3.Constitutive model calibration: Add a sentence stating the criteria for stopping the constant velocity tensile experiments. For instance, was the constant velocity applied until the sample broke?
· Pg.7 - Line 191, 3.Constitutive model calibration: Change "100mm/min" to "100 mm/min", by adding a space between the value and its unit.
· Pg.7 - Line 192, 3.Constitutive model calibration: Change "0.021s^-1" to "0.021 s^-1", by adding a space between the value and its unit.
· Pg.7 - Line 195, 3.Constitutive model calibration: Include a statement as to why cameras in stereo were used for the experiments, as shown by the schematic in Figure 5.
· Pg.7 - Line 200, 3.Constitutive model calibration: Mention that DIC has also been applied to study the viscoelastic behavior of HTPB simulant by citing the following paper https://doi.org/10.1002/prep.202100055, which also captured the rate-dependent damage behavior of the inert HTPB propellant; although at very high strain rates on the order of 10^4/s.
· Pg.8, Figure 6: Include a scale bar for both subfigures. Also, add in the Figure 6 caption that subfigure 6a is showing strain in the tensile direction and include a coordinate system or something to that effect that illustrates which direction is tensile (e.g. horizontal double headed arrow that says tensile direction).
· Pg.8 - Figure 7: Add a space between sigma and the unit (MPa) in vertical axis label.
· Pg.8, Figure 7: Change "stress" to "Stress" in the figure caption by capitalizing. Also, add a space between the values and units in the legend for Figure 7.
· Pg.8, Figure 7: Provide the standard deviation for averaged values (e.g. by using a 95% confidence interval in Figure 7).
· Pg.8 - Line 211, 3.Constitutive model calibration: Delete "obviously".
· Pg.9 - Figure 8: Add a space between sigma and the unit (MPa) in vertical axis label.
· Pg.9 - Line 233, 3.Constitutive model calibration: Change "under different four constant " to "under four different constant".
· Pg.9 - Line 234, 3.Constitutive model calibration: Provide more details about how the model parameters were determined.
· Pg.10, Figure 9: Should the damage variable have units in the figure?
· Pg.10 - Line 244, 3.Constitutive model calibration: Mention if a mesh convergence study was performed, the type of element that was used in ABAQUS, and the total number of elements.
· Pg.11, Figure 11: Add a space between sigma and the unit (MPa) in vertical axis label.
· Pg.11, Figure 12: Add a space between sigma and the unit (MPa) in vertical axis label.
· Pg.12 - Line 270, 3.Constitutive model calibration: Change "inside material, stress" to "inside the material, the".
· Pg.12 - Line 271, 3.Constitutive model calibration: Change "is plotted" to "are plotted".
· Pg.12, Figure 13: Add a space between a value and its unit (e.g. 1 mm/min NOT 1mm/min) in the four subfigure captions. Also, should the damage variable have units on the vertical axis on the right?
· Pg.12, Figure 14: Add the units strain rate in the horizontal axis label. Also, should the damage variable have units on the vertical axis on the left?
· Pg.13 - Line 283, 3.Constitutive model calibration: Change "to define" to "defining".
· Pg.13 - Line 284, 3.Constitutive model calibration: Change "of material" to "of the material".
· Pg.13 - Line 286, 3.Constitutive model calibration: Change "denotes as" to "is denoted as".
· Pg.13 - Line 302, 4.Conclusions: Change "soften" to "softening".
· Pg.13 - Line 308, 4.Conclusions: Change "agrees well" to "agree well".
Author Response
Response to Reviewer 1 Comments
Dear Editor-in-chief and reviewers,
Thank you very much for your letter concerning our manuscript entitled “A nonlinear viscoelastic constitutive model for solid propellant with rate-dependent cumulative damage” (Manuscript ID: materials-1805419). The authors are grateful to the reviewers for their valuable suggestions and comments. We have revised our manuscript in the light of the comments and hope that the improved manuscript is acceptable for publication in Materials. The responses to the comments are listed as follows. Alterations to the original manuscript are marked up using the “Track Changes” function on the revised manuscript. The unmarked manuscript is attached to the marked manuscript.
Sincerely Yours,
Shenghao Chen, Chunguang Wang*, Kaining Zhang, Xuan Lu and Qun Li
E-mail: csh1028@stu.xjtu.edu.cn
wangchunguang@mail.xjtu.edu.cn
knzhang@stu.xjtu.edu.cn
3120306013@stu.xjtu.edu.cn
qunli@mail.xjtu.edu.cn
The authors performed experiments and finite element modeling of hydroxyl-terminated polybutadiene (HTPB) propellants to establish a failure criterion that quantifies the degree of damage in a way that is independent of strain rate. Stress relaxation and constant velocity tensile experiments were performed to calibrate the parameters of the nonlinear viscoelastic model that was developed to study cumulative damage. The finite element simulation modeled a uniaxial tensile test. The theoretical framework for establishing the nonlinear viscoelastic model was provided, along with graphs comparing the results from the simulations with the experiments. The results of this work will be beneficial for those interested in characterizing the nonlinear behavior of composite energetic materials. However, there are a few general comments that need to be addressed:
Comment 1: More details are needed for the experiments and how the constants were calibrated during the fitting process.
Response:
Thanks for your suggestions. The corresponding modifications have been made in the manuscript.
- More details are needed for the experiments
The details for the experiments have been added in the manuscript including the information of HTPB preparation, the reference standard for experimental design and some other additional experimental operation details.
Line 194-197: The material studied in this paper is HTPB propellant, which is produced by curing at 60 ℃ for seven days. The main chemical components include HTPB, AP (ammonium perchlorate), AL (aluminum powder) and the others. The mass fractions are 8%, 69.5%, 18.5% and 4% respectively.
Lines 200-202: In the relaxation test, specimens were stretched at a tensile velocity of 100mm/min to strain level of 5% and kept the strain constant for 2000 s according to the aerospace industry standard of PRC, QJ 2487-1993.
Line 205: Each tensile experiment was carried out until the sample broke.
Line 206-207: The constant velocity tensile experiments were carried out in accordance with the aerospace industry standard of PRC, QJ 924-1985.
- how the constants were calibrated during the fitting process.
The details for how the constants were calibrated during the fitting process have been added in the manuscript. Please see line 171-187.
(1) It is clear from Eq.(1) and Eq.(2) that ER will be eliminated during the calculation of stress. Without loss of generality, select reference modulus ER=1 and take the Prony series expression Eq.(16) for relaxation modulus into Eq.(1) to obtain the pseudo strain εR. That is to say that the εR-t curves can be obtained.
(2) According to the constant velocity tensile experimental results, the σ-t curves can be obtained. Combining the εR-t curves, the C-εR curves can be obtained by Eq.(2).
(3) Assign initial values to damage parameters (a, a, b), and damage internal variable S can be calculated by Eq.(5). Then S-εR relationship can be obtained, and C-S relationship can be ensured.
(4) Plot all C-S curves under different strain rates together and determine whether the overlap ratio is good enough. Then, the values of damage parameters (a, a, b) will be adjusted appropriately until the C-S curves have a good contact ratio, and the damage parameters are what we want.
(5) According to the final C-S curves, determine the form of softening function C(S) and the values of the parameters in the function.
Comment 2: Throughout the paper, make sure to correct small grammatical errors (Detailed and textual comments are as follows:).
Response:
Thanks for your suggestions. According to the detailed and textual comments, the corresponding modifications have been made in the manuscript and will be explained separately.
- 1 - Line26, Abstract: Change "criterion is verified" to "criterion is validated".
It has been corrected.
- 1 - Line 31, 1.Introduction: Change "material consists" to "material and consists".
It has been corrected.
- 1 - Line 34, 1.Introduction: Define "CT" before using the acronym for computed tomography.
It has been corrected.
- 1 - Line 40, 1.Introduction: Change "of solid" to "of a solid".
It has been corrected.
- 1 - Line 41, 1.Introduction: Change "of solid" to "of a solid".
It has been corrected.
- 2 - Line 54, 1.Introduction: Change "method(FEM)" to "method (FEM)", by adding a space after "method".
It has been corrected
- 2 - Line 55, 1.Introduction: Change "verified" to "validated".
It has been corrected.
- 2 - Line 60, 1.Introduction: Change "micro CT" to "micro-CT" with the hypen to be consistent with the notation from Line 34.
It has been corrected.
- 2 - Line 68, 1.Introduction: Change "to related" to "and related".
It has been corrected.
- 2 - Line 68, 1.Introduction: Delete the word "obviously".
It has been corrected.
- 2 - Lines 79-80, 1.Introduction: Reword the sentence that starts with "Section 2 ...", for clarity.
It has been corrected
- 2 - Line 79, 1.Introduction: Change "remaining" to "remainder".
It has been corrected.
- 2 - Line 91, 2.Constitutive model: Define the variable tau in Equation 1.
It has been corrected. The time argument τ is specified as the variable of integration.
- 3 - Line 94, 2.Constitutive model: Change "appear to be primarily" to "appear primarily".
It has been corrected.
- 3 - Line 101, 2.Constitutive model: Define the variable sigma (stress) in Equation 2.
It has been corrected. σ is the stress tensor.
- 3 - Line 105, 2.Constitutive model: Change "describe mechanical" to "describe the mechanical".
It has been corrected.
- 3 - Line 107, 2.Constitutive model: Change "is classical" to "is a classical".
It has been corrected.
- 3 - Line 112, 2.Constitutive model: Change "dependence(see" to "dependence (see" by adding a space before the open parenthesis.
It has been corrected.
- 3 - Lines 114-115, 2.Constitutive model: Change "is same as stress-time" to "is the same as the stress-time".
It has been corrected.
- 3 - Line 115, 2.Constitutive model: Change "is power" to "is the power".
It has been corrected.
- 3 - Line 116, 2.Constitutive model: Change "function, as" to "function, defined as".
It has been corrected.
- 3 - Line 128, 2.Constitutive model: Change "is in detailed in" to "is detailed in".
It has been corrected.
- 4 - Line 131, 2.Constitutive model: Add another sentence describing the material and experiment conditions used to generate Figure 1. Otherwise, remove the values on the horizonal and vertical axis in Figure 1 to make the plot generic like Figure 2 is generic.
The values on the horizontal and vertical axis in Figure 1 have been removed to make the plot generic.
- 5 - Lines 141-142, 2.Constitutive model: Add a citation to support the statement that "the ramp time is large."
A citation has been added to support the statement that “the ramp time is large”.
- 5 - Line 142, 2.Constitutive model: Change "acquire precise" to "acquire the precise".
It has been corrected.
- 5 - Line 143, 2.Constitutive model: Change "Substitute" to "Substituting".
It has been corrected.
- 5 - Line 153, 2.Constitutive model: Change "is the form of Prony" to "is in the form of a Prony".
It has been corrected.
- 5 - Line 157, 2.Constitutive model: Provide more details for how the constants from Equation 16 were obtained. For instance, what program was used and what function in the program was used for optimizing the constants?
It has been corrected. Each coefficient of Eq.(16) can be obtained by the Nonlinear Least Squares Method, which is implemented by the fit function of MATLAB software.
- 7 - Line 185, 3.Constitutive model calibration: Provide more details on how the HTPB propellant was made (e.g. curing temperature and curing time, type of oxidizer, weight fraction of the components, etc.).
It has been corrected. The material studied in this paper is HTPB propellant, which is produced by curing at 60 ℃ for seven days. The main chemical components include HTPB, AP (ammonium perchlorate), AL (aluminum powder) and the others. The mass fractions are 8%, 69.5%, 18.5% and 4% respectively.
- 7 - Line 187, 3.Constitutive model calibration: Change "tensile mechanic properties" to "tensile mechanical properties".
It has been corrected.
- 7 - Line 189, 3.Constitutive model calibration: Change "100mm/min" to "100 mm/min", by adding a space between the value and its unit.
It has been corrected.
- 7 - Lines 189-190, 3.Constitutive model calibration: Add another sentence or two explaining why the chosen tensile velocity of 100 mm/min and strain level of 5% was used for the stress relaxation experiments. 5% strain is small and is not expected to cause large deformations in the sample resulting in the need for a nonlinear viscoelastic model.
It has been corrected. “In the relaxation test, the chosen tensile velocity of 100 mm/min and strain level of 5% was used according to the aerospace industry standard of PRC, QJ 2487-1993.”
Relaxation test is the most common approach to determine relaxation properties of viscoelastic materials. The test should be carried out without causing material damage. That is why small strains (i.e. 5%) are often used in relaxation test. In the uniaxial tensile test in this research, the strain is above 15%, which is used to study the damage behavior of the material.
- 7 - Line 190, 3.Constitutive model calibration: Change "2000s" to "2000 s", by adding a space between the value and its unit.
It has been corrected.
- 7 - Line 190, 3.Constitutive model calibration: Add a sentence stating the criteria for stopping the constant velocity tensile experiments. For instance, was the constant velocity applied until the sample broke?
It has been corrected. Each tensile experiment was carried out until the sample broke
- 7 - Line 191, 3.Constitutive model calibration: Change "100mm/min" to "100 mm/min", by adding a space between the value and its unit.
It has been corrected.
- 7 - Line 192, 3.Constitutive model calibration: Change "0.021s^-1" to "0.021 s^-1", by adding a space between the value and its unit.
It has been corrected.
- 7 - Line 195, 3.Constitutive model calibration: Include a statement as to why cameras in stereo were used for the experiments, as shown by the schematic in Figure 5.
It has been corrected. Some information about DIC equipment, the cameras and lens have been added in the manuscript to explain the schematic in Figure 5. “Experimental facilities included Zwick/Roell-Z005 type universal testing machine, DIC technique equipment (VIC-3D, Beijing Ruituo Technology Co., Ltd. China), and two cameras with a resolution of 2048 × 2048, with 50 mm macro lens attached.”
- 7 - Line 200, 3.Constitutive model calibration: Mention that DIC has also been applied to study the viscoelastic behavior of HTPB simulant by citing the following paper https://doi.org/10.1002/prep.202100055, which also captured the rate-dependent damage behavior of the inert HTPB propellant; although at very high strain rates on the order of 10^4/s.
The paper (https://doi.org/10.1002/prep.202100055) has been cited in the manuscript. What’s more, the technique has also been applied to study the viscoelastic behavior of HTPB [34].
- 8, Figure 6: Include a scale bar for both subfigures. Also, add in the Figure 6 caption that subfigure 6a is showing strain in the tensile direction and include a coordinate system or something to that effect that illustrates which direction is tensile (e.g. horizontal double headed arrow that says tensile direction).
It has been corrected. The scale bar of Figure 6b and the horizontal double arrow that says tensile direction have been added in Figure 6.
- 8 - Figure 7: Add a space between sigma and the unit (MPa) in vertical axis label.
It has been corrected.
- 8, Figure 7: Change "stress" to "Stress" in the figure caption by capitalizing. Also, add a space between the values and units in the legend for Figure 7.
It has been corrected.
- 8, Figure 7: Provide the standard deviation for averaged values (e.g. by using a 95% confidence interval in Figure 7).
The standard deviation for averaged values has been provided in line 206-207. “The standard deviations for averaged values are within 2%.”
The experimental data with tensile velocity of 20 mm/min is taken as an example to illustrate in the cover letter. The figure(please see the attachment) shows the average curve and associated error band diagram. The maximum standard deviation is 1.3%.
- 8 - Line 211, 3.Constitutive model calibration: Delete "obviously".
It has been corrected.
- 9 - Figure 8: Add a space between sigma and the unit (MPa) in vertical axis label.
It has been corrected.
- 9 - Line 233, 3.Constitutive model calibration: Change "under different four constant " to "under four different constant".
It has been corrected.
- 9 - Line 234, 3.Constitutive model calibration: Provide more details about how the model parameters were determined.
It has been corrected. Based on the stress-time data of relaxation test, the stress rate can be obtained by Eq.(15). Substitute the stress rate into Eq.(14) to obtain the value of relaxation modulus. The expression of relaxation modulus can be obtained by the Nonlinear Least Squares Method, which is implemented by MATLAB software.
- 10, Figure 9: Should the damage variable have units in the figure?
The damage variable S has no units as shown in Eq.4. Therefore, the damage variable on the vertical or horizontal axis is not labeled with units.
- 10 - Line 244, 3.Constitutive model calibration: Mention if a mesh convergence study was performed, the type of element that was used in ABAQUS, and the total number of elements.
It has been corrected. The finite element model is shown in Figure 10 and its dimension is 50 mm × 10 mm × 10 mm. The finite element mesh applied in this case consists of eight-node 3D stress elements with reduced integration. The mesh size is controlled by a global size of 0.5 mm, and the number of elements is 40000.
- 11, Figure 11: Add a space between sigma and the unit (MPa) in vertical axis label.
It has been corrected.
- 11, Figure 12: Add a space between sigma and the unit (MPa) in vertical axis label.
It has been corrected.
- 12 - Line 270, 3.Constitutive model calibration: Change "inside material, stress" to "inside the material, the".
It has been corrected.
- 12 - Line 271, 3.Constitutive model calibration: Change "is plotted" to "are plotted".
It has been corrected.
- 12, Figure 13: Add a space between a value and its unit (e.g. 1 mm/min NOT 1mm/min) in the four subfigure captions. Also, should the damage variable have units on the vertical axis on the right?
It has been corrected.
- 12, Figure 14: Add the units strain rate in the horizontal axis label. Also, should the damage variable have units on the vertical axis on the left?
It has been corrected.
- 13 - Line 283, 3.Constitutive model calibration: Change "to define" to "defining".
It has been corrected.
- 13 - Line 284, 3.Constitutive model calibration: Change "of material" to "of the material".
It has been corrected.
- 13 - Line 286, 3.Constitutive model calibration: Change "denotes as" to "is denoted as".
It has been corrected.
- 13 - Line 302, 4.Conclusions: Change "soften" to "softening".
It has been corrected.
- 13 - Line 308, 4.Conclusions: Change "agrees well" to "agree well".
It has been corrected.
Comment 3: Expand the Section titled "3.1. Uniaxial tensile tests and relaxation tests of HTPB " by including details of the DIC experiments. Details should include (i) the cameras and lens were used, the (ii) calibration grid size, (iii) the software used for DIC, and (iv) other items detailed in the DIC guide found at https://idics.org/guide/. Make sure to cite the guide.
Response:
Thanks for your suggestion. Some details have been added in the manuscript.
Lines 209-211:
Experimental facilities included Zwick/Roell-Z005 type universal testing machine, DIC technique equipment (VIC-3D, Beijing Ruituo Technology Co., Ltd. China), and two cameras with a resolution of 2048 × 2048, with 50 mm macro lens attached.
Lines 222-224:
The surface information of HTPB was quantified using the software VIC-3D [35]. A standard calibration target that had 14 dots × 10 dots, with a dot spacing of 14 mm was used during the calibration process.
Comment 4: The authors should include a figure (or table) that applies Equation 21 so the reader can see the damage degree of the samples from the experiments performed in this study. A statement of how the damage degree from Equation 21 was validated experimentally should be included (e.g. via CT or SEM).
Response:
Thanks for your suggestion. We agree to your comments. We added a Figure 15 so that the reader can see the curves of the damage degree at different tensile velocities. Please see lines 320-324 and Figure 15.
“Figure 15 shows the evolution of damage degree as it relates to the experimental data at different tensile velocities. As shown in Figure 15, the condition`S=0 defines a material state wherein there is no damage, and the condition`S=1 defines the state of failure in the material with the maximum stress attained.” It would be better if the damage degree from Equation 21 was validated experimentally through CT or SEM. This verification work was not carried out due to the limitations of samples and experiment.

Reviewer 2 Report
The finite element model is not properly described. The description should be extended by providing the information about the dimensions of the model, the element type, the mesh, the material models, etc.
Author Response
Response to Reviewer 2 Comments
Dear Editor-in-chief and reviewers,
Thank you very much for your letter concerning our manuscript entitled “A nonlinear viscoelastic constitutive model for solid propellant with rate-dependent cumulative damage” (Manuscript ID: materials-1805419). The authors are grateful to the reviewers for their valuable suggestions and comments. We have revised our manuscript in the light of the comments and hope that the improved manuscript is acceptable for publication in Materials. The responses to the comments are listed as follows. Alterations to the original manuscript are marked up using the “Track Changes” function on the revised manuscript. The unmarked manuscript is attached to the marked manuscript.
Sincerely Yours,
Shenghao Chen, Chunguang Wang*, Kaining Zhang, Xuan Lu and Qun Li
E-mail: csh1028@stu.xjtu.edu.cn
wangchunguang@mail.xjtu.edu.cn
knzhang@stu.xjtu.edu.cn
3120306013@stu.xjtu.edu.cn
qunli@mail.xjtu.edu.cn
Comment 1: The finite element model is not properly described. The description should be extended by providing the information about the dimensions of the model, the element type, the mesh, the material models, etc.
Response:
Thanks for your suggestion. The description of the finite element model has been extended in the manuscript. Please see lines 268-269 and 272-274.
The finite element model is shown in Figure 10 and its dimension is 50 mm × 10 mm × 10 mm. The finite element mesh applied in this case consists of eight-node 3D stress elements with reduced integration. The mesh size is controlled by a global size of 0.5 mm, and the number of elements is 40000.

Reviewer 3 Report
I have following concerns/suggestions regarding the paper.
Line 40-write "solid propellant rocket " instead of "solid rocket".
Line 40-41 is not essential since it is not part of the current research. Line 67-68 is unclear and should be rephrased. Line 79-84 is not necessary and such write up is usually found in thesis or project reports and not required in a journal.
Line 93-94 is unclear and should be re-written properly.
Line 108 should be written as " stress-time loading history, is given by"
Line 109, how are "k and beta" determined? What do they signify? It is not clear.
Line 128-129 Write it as " which is expressed as"
Figure 1- It is not clear whether the authors have experimentally plotted? if not it should be cited appropriately.
Section 3.1- was the tensile test done in accordance with ASTM standards or ISO standard? Authors have mentioned that tests were repeated 3 times. How many samples were tested and what was the degree of repeatability?
Rephrase sentence in line 201-202.
Authors mention that the behavior is non-linear based on the plot in figure 7. But it looks like the behavior is almost linear up to a strain of 0.15 and after that it is nonlinear. Clarify and explain the phenomenon.
How is equation 21 derived and what is the basis for it? The authors have to prove its validity.
Some details about the HTPB is missing and about the sample preparation. Also, the assumptions related to the modeling is also missing and should be appropriately addressed.
Author Response
Response to Reviewer 3 Comments
Dear Editor-in-chief and reviewers,
Thank you very much for your letter concerning our manuscript entitled “A nonlinear viscoelastic constitutive model for solid propellant with rate-dependent cumulative damage” (Manuscript ID: materials-1805419). The authors are grateful to the reviewers for their valuable suggestions and comments. We have revised our manuscript in the light of the comments and hope that the improved manuscript is acceptable for publication in Materials. The responses to the comments are listed as follows. Alterations to the original manuscript are marked up using the “Track Changes” function on the revised manuscript. The unmarked manuscript is attached to the marked manuscript.
Sincerely Yours,
Shenghao Chen, Chunguang Wang*, Kaining Zhang, Xuan Lu and Qun Li
E-mail: csh1028@stu.xjtu.edu.cn
wangchunguang@mail.xjtu.edu.cn
knzhang@stu.xjtu.edu.cn
3120306013@stu.xjtu.edu.cn
qunli@mail.xjtu.edu.cn
Comment 1: Line 40-write "solid propellant rocket " instead of "solid rocket".
Response:
Thanks for your suggestion. It has been corrected.
Comment 2: Line 40-41 is not essential since it is not part of the current research. Line 67-68 is unclear and should be rephrased. Line 79-84 is not necessary and such write up is usually found in thesis or project reports and not required in a journal.
Response:
Thanks for your suggestions.
Line 40-41 has been deleted.
Line 67-68 has been rephrased. The yield stress and failure stress are observed experimentally and related to the tensile velocity.
Line 79-84 has been deleted.
Comment 3: Line 93-94 is unclear and should be re-written properly.
Response:
Thanks for your suggestion. The corresponding modifications have been made in the manuscript.
The initiation and growth of microcracks and microvoids occur under loading. During this process, damage behavior appears primarily which leads to a nonlinear material behavior.
Comment 4: Line 108 should be written as " stress-time loading history, is given by"
Response:
Thanks for your suggestion. It has been corrected.
Comment 5: Line 109, how are "k and beta" determined? What do they signify? It is not clear.
Response:
Thanks for your comment. The function (Eq.4) is from Ref [29], and a nonlinear viscoelastic constitutive equation incorporating cumulative damage in developed in the Ref [29]. The fitting parameters k and beta of Eq.(4) are characteristic cumulative damage parameters of the material. The values of the two parameters can be determined by tensile test results.
The corresponding modifications have been made in the manuscript. Please see line 113-114.
“where k and b are characteristic cumulative damage parameters of the material. They can be obtained from tensile test results at different strain rates.”
Comment 6: Line 128-129 Write it as " which is expressed as"
Response:
Thanks for your suggestion. It has been corrected.
Comment 7: Figure 1- It is not clear whether the authors have experimentally plotted? if not it should be cited appropriately.
Response:
Thanks for your comment. The figure is plotted based on our own experiment. In order to make the plot generic, the values on the horizontal and vertical axis in Figure 1 have been removed. Please see Figure 1.
Comment 8: Section 3.1- was the tensile test done in accordance with ASTM standards or ISO standard? Authors have mentioned that tests were repeated 3 times. How many samples were tested and what was the degree of repeatability?
Response:
Thanks for your comment. The corresponding modifications have been made in the manuscript.
- was the tensile test done in accordance with ASTM standards or ISO standard?
“The constant velocity tensile experiments were carried out in accordance with the aerospace industry standard of PRC, QJ 924-1985.”
Please see line 207-208.
- Authors have mentioned that tests were repeated 3 times. How many samples were tested and what was the degree of repeatability?
15 samples were tested and the standard deviations for averaged values are within 2%.
The experimental data with tensile velocity of 20 mm/min is taken as an example to illustrate in the cover letter. The figure (please see the attachment) shows the average curve and associated error band diagram. The maximum standard deviation is 1.3%.
Comment 9: Rephrase sentence in line 201-202.
Response:
Thanks for your suggestion. The corresponding modification has been made in the manuscript. Please see line 219-221.
“In order to increase accuracy of measurement, speckle patterns are usually applied by spraying the specimen with contrasting paints (i.e. black, white and gray).”
Comment 10: Authors mention that the behavior is non-linear based on the plot in figure 7. But it looks like the behavior is almost linear up to a strain of 0.15 and after that it is nonlinear. Clarify and explain the phenomenon.
Response:
Thanks for your comment. As mentioned in Introduction, propellant is a complex particulate composite material and consists of relatively stiff solid particles embedded in a matrix of soft polymeric material. We think the internal bonding performance of propellant is good and there are no defects in the initial stage of loading. Hence, the behavior is almost linear. After that, microflaws initiate and grow at the matrix and the particle/matrix interface, which cause the particle/matrix interface debonding. This is the main reason for the nonlinearity. During the loading process, the damage inside the propellant accumulates gradually. It should be noted that the damage variable defined in this paper can explain the change from linear to nonlinear, as shown in Figure 13. When the strain is in a small range, the value of damage variable S is also small. With the increase of strain, the value of S gradually increases, and the material exhibits nonlinearity.
Comment 11: How is equation 21 derived and what is the basis for it? The authors have to prove its validity.
Response: Thanks for your comment. We agree to your comment.
- How is equation 21 derived and what is the basis for it?
As mentioned in Section 3.3, the parameter`S in equation 21 is self-defined failure criterion in this research. The advantage of this approach is that`S value at different strain-rates is within 1 after normalization, which means that`S is independent of strain rate. We found that the relationship between Sm (the value of damage variable S at failure point) and strain rate is linear, which is expressed as Equation 20. In order to define a new failure criterion which is independent of strain rate, the normalization operation was performed. The new failure criterion is defined as the ratio of S and Sm , denoted as`S.
- The authors have to prove its validity.
In order to prove its validity, we added a Figure 15. The corresponding modifications and supplements have been made in the manuscript. Please see Figure 15 and line 320-324.
“Figure 15 shows the evolution of damage degree as it relates to the experimental data at different tensile velocities. As shown in Figure 15, the condition`S=0 defines a material state wherein there is no damage, and the condition`S=1 defines the state of failure in the material with the maximum stress attained.”
Comment 12: Some details about the HTPB is missing and about the sample preparation. Also, the assumptions related to the modeling is also missing and should be appropriately addressed.
Response: Thanks for your suggestion.
- Some details about the HTPB is missing and about the sample preparation.
The more details about the sample preparation (curing temperature and curing time, the main components, weight fraction of the components) have been provided. Please see line 194-197.
“The material studied in this paper is HTPB propellant, which is produced by curing at 60 ℃ for seven days. The main chemical components include HTPB, AP (ammonium perchlorate), AL (aluminum powder) and the others. The mass fractions are 8%, 69.5%, 18.5% and 4% respectively.”
- Also, the assumptions related to the modeling is also missing and should be appropriately addressed.
The description of the finite element model has been extended. Please see line 269-270.
“The specimen is simplified to a rectangular geometry based on the assumption that the stress in the cross section in uniformly distributed under tensile loading.”

Round 2
Reviewer 1 Report
The revised manuscript is greatly improved from its original version. Many issues have been clarified. The reviewer does not have any additional comments or questions regarding the revised manuscript.